# Triplex-Loop-Mediated Isothermal Amplification Combined with a Lateral Flow Immunoassay for the Simultaneous Detection of Three Pathogens of Porcine Viral Diarrhea Syndrome in Swine

**DOI:** 10.3390/ani13121910

**Published:** 2023-06-07

**Authors:** Yi Hong, Biao Ma, Jiali Li, Jiangbing Shuai, Xiaofeng Zhang, Hanyue Xu, Mingzhou Zhang

**Affiliations:** 1Zhejiang Provincial Key Laboratory of Biometrology and Inspection & Quarantine, China Jiliang University, Hangzhou 310018, China; 18768152453@163.com (Y.H.); 16a0701109@cjlu.edu.cn (B.M.); 2Hangzhou Quickgene Sci-Tech. Co., Ltd., Hangzhou 310018, China; qjc1993@126.com; 3Zhejiang Academy of Science and Technology for Inspection and Quarantine, Hangzhou 310016, China; sjb@zaiq.org.cn (J.S.);; 4College of Life Science, China Jiliang University, Hangzhou 310018, China; m19357389472@163.com

**Keywords:** loop-mediated isothermal amplification, lateral flow dipstick, multiple detection, porcine viral diarrhea, in-field

## Abstract

**Simple Summary:**

Porcine epidemic diarrhea virus (PEDV), porcine bocavirus (PBoV), and porcine rotavirus (PoRV) are associated with porcine viral diarrhea. The three viruses are often comorbid in mixed infection; therefore, it is necessary to develop a specific and rapid method for simultaneous detection. In this study, a multiple detection method, loop-mediated isothermal amplification (LAMP) combined with a lateral flow dipstick (LFD), was established to detect three pathogens in 125 field samples. The consistency between the real-time fluorescence quantitative PCR (rt-qPCR) and the triplex LAMP–LFD assay was over 99%. The assay has been proven to be highly specific and sensitive.

**Abstract:**

Porcine epidemic diarrhea virus (PEDV), porcine bocavirus (PBoV), and porcine rotavirus (PoRV) are associated with porcine viral diarrhea. In this study, triplex loop-mediated isothermal amplification (LAMP) combined with a lateral flow dipstick (LFD) was established for the simultaneous detection of PEDV, PoRV, and PBoV. The PEDV-*gp6*, PoRV-*vp6*, and PBoV-*vp1* genes were selected to design LAMP primers. The amplification could be carried out at 64 °C using a miniature metal bath within 30 min. The triplex LAMP–LFD assay exhibited no cross-reactions with other porcine pathogens. The limits of detection (LODs) of PEDV, PoRV, and PBoV were 2.40 × 10^1^ copies/μL, 2.89 × 10^1^ copies/μL, and 2.52 × 10^1^ copies/μL, respectively. The consistency between rt-qPCR and the triplex LAMP–LFD was over 99% in field samples testing. In general, the triplex LAMP–LFD assay was suitable for the rapid and simultaneous detection of the three viruses in the field.

## 1. Introduction

For a long time, pork has been the most important meat product and the main source of animal protein supplementation in daily life. With the expansion of the scale of pig breeding, the prevention and control of porcine diseases is one of the biggest problems in the breeding industry. Porcine viral diarrhea is a key contemporary cause of high morbidity and high mortality in piglets, causing great economic losses and major public health concerns around the world [1]. In the winter of 2010, a violent outbreak of PED occurred on pig farms in southern China, resulting in the deaths of more than one million piglets. It then rapidly spread across the country [2]. In 2013, a PEDV variant caused a pandemic in the United States and circulated to Canada and Mexico. It was estimated that the U.S. lost nearly 7 million pigs between 2013 and 2015, causing devastating damage to the pig industry [3]. In addition to the direct losses from piglet deaths, the indirect economic losses, including disease treatment and prevention, cannot be ignored. The higher incidence of disease incurs additional costs for pig farms. To ensure the stability of the pig breeding industry, it is essential to develop an accurate diagnostic method for swine diarrhea virus.

Porcine epidemic diarrhea virus (PEDV), porcine bocavirus (PBoV), and porcine rotavirus (PoRV) are associated with porcine viral diarrhea, which can lead to severe diarrhea, vomiting, dehydration and decreased appetite in infected pigs [4,5]. PEDV was first identified as the causative agent of porcine epidemic diarrhea (PED) in 1978 in Europe [6]. PEDV infection is currently widespread in pig-farming countries in Asia, Europe and North America [7,8,9]. PEDV is a single-stranded RNA virus, which belongs to the *Alpha coronavirus* [10]. As a structural protein, the nucleocapsid protein plays an important role in viral core formation, viral assembly and the cell stress response [11,12,13]. The *gp6* gene, which can encode the nucleocapsid protein, has been used as a target for PEDV detection and treatment [14,15]. PBoV was first identified in 2009 in Swedish pigs with the serious infectious disease, post-weaning multisystemic wasting syndrome (PMWS). The prevalence of PBoV was 46% in pigs without PMWS, and 88% in pigs with PMWS [5,16,17,18]. Within a year, PBoV was observed in diarrheic pigs, as well as pigs suffering from respiratory problems [5]. It has subsequently been reported worldwide [5,19,20,21,22]. PBoV is a non-enveloped single-stranded DNA virus belonging to the *Boca virus genus* of the *Parvoviridae subfamily parvoviridae* [23]. The genome of PBoV consists of three open reading frames (ORFs), encoding the NS1, NP1, VP1, and VP2 proteins [24]. The *vp1* gene was selected as the fragment for identifying PBoV [20,25]. PoRV was first isolated from infected pigs in 1976 [26], and has since become one of the major factors of neonatal diarrhea in swine worldwide [27]. PoRV is a double-stranded RNA virus belonging to *Rotavirus*. The *vp4* gene is usually utilized to analyze the biological characteristics of PoRV [28]. The VP6 protein, which is highly immunogenic and antigenic, is also used for the detection of PoRV [29]. Some studies have indicated that diarrhea in piglets is associated with the co-infection of PEDV, PoRV, and PBoV [4,30,31,32].

Laboratory diagnostic methods, which are commonly used for porcine viral diarrhea detection, include the virus isolation method [33], enzyme-linked immunoadsorption assays (ELISAs) [34], indirect immunofluorescence assays (IFAs) [35], neutralization assays (NTs) [36], and real-time fluorescence quantitative PCR (rt-qPCR) [37]. The virus isolation method is time-consuming and complicated [38]. ELISA, IFA, NTs, and other serological methods require the preparation of the antigen or antibody in advance [39]. Real-time fluorescent quantitative PCR has relatively good specificity and sensitivity. However, the dependency on equipment limits its application in testing samples in the field [38,39].

Loop-mediated isothermal amplification (LAMP) is a nucleic acid isothermal amplification technique, which can be carried out at a constant temperature using *Bst* DNA polymerase [40,41]. The assay uses four primers to identify six specific regions of the target DNA [42]. It can amplify target fragments exponentially within 1 h at 60–65 °C. Compared with typical PCR, it does not rely on special instruments: a dry block heater or an incubator can be used [43]. After amplification, the products are usually visualized through turbidimetry [44], agarose gel electrophoresis [45], fluorescent dye [46], and fluorescent probes [47]. The turbidimetry and fluorescent dye methods cannot detect multiple targets simultaneously. The agarose gel electrophoresis method has difficulties in meeting the conditions of testing outside the laboratory. The fluorescent probe methods can monitor multiple targets at the same time [38]. However, the assays depend on special fluorescence quantitative instruments. The lateral flow dipstick (LFD) is a simple, rapid and visual method. The combination of LAMP and LFD had been proven to be effective and compatible with on-site testing [48,49].

In this study, a triplex LAMP–LFD assay was developed to detect PEDV, PoRV, and PBoV simultaneously. It could be completed within 30 min in a MiniT-100H metal bath (Allsheng Instruments Co. Ltd., Hangzhou, China). Based on the visualization, the test strip reader (GIC-S100-B14, Suzhou Helmen Precise Instruments, Suzhou, China) was employed to scan and evaluate the colored signals on the lateral flow dipsticks for quantitative analyses. In conclusion, the triplex LAMP–LFD assay established in this study is a convenient, sensitive and specific technique for testing samples in the field. The triplex LAMP–LFD assay is expected to make a significant contribution to clinical detection in the future.

## 2. Materials and Methods

### 2.1. Virus Strains and Viral Genes

PEDV (CV777 strain), PoRV (NX strain), PBoV (CH437 strain), Porcine circovirus type 1 (PCV1, NJ03 strain), Transmissible gastroenteritis virus (TGEV, H strain), Porcine reproductive and respiratory syndrome virus (PRRSV, TJM-F92 strain), Pseudorabies virus (PRV, Bartha-K61 strain), and Porcine parvovirus (PPV, S-2 strain) were provided by Zhejiang Academy of Science and Technology for Inspection and Quarantine (Hangzhou, China). The virus strains propagated in susceptible cells [50]. According to the manufacturer’s instructions, the viral DNA and RNA were extracted with the DNA/RNA Isolation Kit (TIANGEN, Beijing, China). The viral RNA was reverse-transcribed with the TIANScriptIIRT Kit (TIANGEN, Beijing, China). The DNA/cDNA was stored at −80 °C until use.

### 2.2. Design of LAMP Primers

Based on the highly conservative *gp6* gene (GenBank accession: NC_003436.1) of PEDV, the *vp6* gene (GenBank accession: KC113249.1) of PoRV and the *vp1* gene (GenBank accession: NC_023673.1) of PBoV, the specific LAMP primers were designed using the online website PrimerExplorer v5 (http://primerexplorer.jp/lampv5e/index.html, accessed on 12 October 2022). The pairs of loop primers, LB and LF, were designed to enhance efficiency and increase specificity. In order to distinguish different amplification products, loop primers with target-specific labels were tagged with biotin and digoxin (loop primers for PEDV), biotin and ROX (loop primers for PoRV) and, biotin and Cy5 (loop primers for PBoV), respectively. All primers (Table 1) were synthesized by Sangon Biotech (Shanghai, China) Co., Ltd.

### 2.3. Preparation of Standard Plasmids

The primers of PEDV F3/B3, PoRV F3/B3, and PBoV F3/B3 were used for PCR amplification. The amplified PCR products were collected and purified using the DNA Gel Recovery Kit (UE-GX-250, UElandy Co., Ltd., Suzhou, China). The target fragments were ligated into the pMD18-T vector (TaKaRa). The recombinant plasmids were transformed into competent Escherichia coli *DH5a*. The positive clones were isolated and identified by sequencing. The plasmids were extracted using the Plasmid Preparation Kit (UE-MN-P-250, UElandy Co., Ltd., Suzhou, China) as standard plasmids. The concentrations were measured using a Nano-100 micro spectrophotometer (Allsheng Instruments Co. Ltd., Hangzhou, China). The copy numbers were calculated based on the following formula: number of copies = (concentration in ng) × 10^−9^ × 6.02 × 10^23^)/(genome length × 660). The standard plasmids were stored at −20 °C until use.

### 2.4. Preparation of the LFD

The lateral flow test strip consists of a sample pad, a binding pad, an NC membrane, and an absorption pad. The colloidal gold solution was labeled with anti-biotin monoclonal antibody and sprayed evenly on the binding pad. Line C was coated with 2 mg/mL goat-anti-mouse polyclonal antibody; the T1 line was coated with 0.75 mg/mL anti-digoxin monoclonal antibody for the detection of PEDV; the T2 line was coated with 0.6 mg/mL anti-ROX monoclonal antibody for the detection of PoRV; and the T3 line was coated with 0.5 mg/mL anti-Cy5 monoclonal antibody for the detection of PBoV. The test paper was assembled, and it was cut into 2.5 mm strips for storage in a dry environment until use.

### 2.5. Construction of the Triplex LAMP–LFD Reaction System

The total volume of the triplex LAMP–LFD reaction system was 25 μL, including 10 × buffer, dNTPs, *Bst* DNA polymerase, primer mixtures and templates. The primer mixtures contained 0.2 μM F3/B3 (PEDV-F3/B3, PoRV-F3/B3, PBoV-F3/B3), 1.6 μM FIP/BIP (PEDV-FIP/BIP, PoRV-FIP/BIP, PBoV-FIP/BIP) and 0.8 μM LF/LB (PEDV-LF/LB, PoRV-LF/LB, PBoV-LF/LB). ddH_2_O was used as a negative control. The reaction was conducted at 64 °C for 40 min. After amplification, the products were diluted 50 times and dropped onto the sample pad for testing. Then, the color signals on the strips were read for quantitative analyses.

### 2.6. Optimization of the Triplex LAMP–LFD Assay

The optimization of the triplex LAMP–LFD assay was performed by using the standard plasmids with 10^5^ copies. The primer mixtures comprised 0.4 μM F3 and B3, 3.2 μM FIP and BIP and 1.6 μM LF and LB. Six primer ratios were tested to determine those of high efficiency, and the proportions of the PEDV, PoRV, and PBoV primers were set as 1.4:1:0.6, 1.2:1:0.8, 1.1:1:0.9, 0.9:1:1.1, 0.8:1:1.2, and 0.6:1:1.4. On this basis, different concentrations of the enzyme were compared, ranging from 0.16 U to 0.96 U. Subsequently, different concentrations of dNTPs were optimized, ranging from 0.8 to 1.8 mM. In addition, the amplification distinctions at different temperatures of 55 °C, 58 °C, 61 °C, 64 °C, 67 °C, and 70 °C were examined. The reactions were carried out from 5 min to 40 min to observe the differences in amplification results using the testing strips. The colored signals were collected and analyzed based on the test strip reader. All experiments were repeated three times.

### 2.7. Evaluation of Specificity and Sensitivity

The specificity of the triplex LAMP–LFD assay was assessed with other porcine pathogens, including PCV1, TGEV, PRRSV, PRV, and PPV. In the sensitivity testing, the standard plasmids were 10-fold serially diluted from 10^6^ copies/μL to 10^0^ copies/μL. The three plasmids at the same concentration level were mixed in an equal volume. Each testing strip was scanned independently three times.

### 2.8. Real-Time PCR Assay Based on Fluorescent Probes

Real-time PCR assay based on fluorescent probes (rt-qPCR) is a relatively common detection method in molecular biology, which is often selected for detecting porcine diarrhea pathogens. The primers and probes of the rt-qPCR assay were designed using Beacon Designer 7.9 software (PREMIER Biosoft, San Francisco, CA, USA). All primers and probes (Appendix A) were synthesized by Sangon Biotech (Shanghai) Co., Ltd. The rt-qPCR reaction system includes 10 μL 2 × Premix Ex *Taq* (Probe qPCR, TaKaRa), 0.2 μL RoxII, 0.8 μM primers sets, 0.1 μM probes, 2.5 μL templates, and ddH_2_O. The rt-qPCR reactions were performed with the following steps: (1) 95 °C for 5 min; (2) 95 °C for 10 s and 60 °C for 30 s, repeating for 40 cycles.

### 2.9. Detection of Field Samples

Between November and December 2022, 125 animal feces samples, including watery feces, were collected from a pig farm in Zhejiang province Appendix A. The anal swab was inserted into the pig anus and slowly rotated. The anal swabs were extracted and stored in the buffer solution. The liquid samples were stored at −20 °C until use. According to the instructions of the FastPure Viral DNA/RNA Mini Kit (Vazyme Biotech Co., Ltd., Nanjing, China), the nucleic acids were quickly extracted from liquid samples. All samples were detected by the triplex LAMP–LFD assay and the rt-qPCR assay. The reactions of the rt-qPCR assay were conducted using QuantStudio™ 5 real-time fluorescence quantitative PCR equipment (Thermo Fisher Scientific Inc., Waltham, MA, USA).

### 2.10. Data Analyses

The T/C value and the logarithm of the copy number were used to draw the standard curves for the triplex LAMP–LFD assay. The data collected from the two assays, were analyzed using QuantStudio™ real-time PCR software v1.5.1 (Thermo Fisher Scientific Inc., Waltham, MA, USA) and Microsoft Excel software 2021 (Microsoft Inc., Redmond, WA, USA). The results of the rt-qPCR assay were judged as positive when the Ct value ≤ 35.

## 3. Results

### 3.1. Assay Principle

The schematic diagram of the triplex LAMP–LFD assay is shown in Figure 1. Three sets of primers were used in the triplex LAMP–LFD reaction system, and the loop primers of each group were labeled with biotin and digoxin (loop primers for PEDV), biotin and ROX (loop primers for PoRV) and biotin and Cy5 (loop primers for PBoV), respectively. The amplicons with different fluorescent groups accumulated along with the amplification. The products with different labels were captured by the corresponding specific antibodies and fixed on the different T-lines, where the anti-digoxin monoclonal antibody was coated on the T1 line, the anti-ROX monoclonal antibody was coated on the T2 line and the anti-Cy5 monoclonal antibody was coated on the T3 line. The non-captured gold particles were immobilized by the secondary antibody on the control line. The experimental results could be identified by the reddish bands. In the presence of the targets, the red bands could be observed on the corresponding test lines. In contrast, no red bands appeared.

### 3.2. Optimization of the Triplex LAMP–LFD Conditions

The T/C value was defined as the test line strength by using the test strip reader for quantitative analyses. When the proportions of the PEDV, PoRV, and PBoV primers were 1.1:1:0.9, the chromaticity of the three T lines was bright and homogeneous, and the T/C values were similar (Figure 2A).

Aiming for a high amplification efficiency, the effect of the *Bst* DNA polymerase concentration was tested in Figure 2B. When the amount of the enzyme was insufficient, such as 0.16 U/μL or 0.32 U/μL, the color development of the T1 line was significantly higher than that of the other two lines, which was judged by naked eyes. Additionally, the number of products did not increase obviously with the increase in enzyme contents after reaching the plateau. According to the data measured through the test strip reader, an enzyme concentration of 0.48 U/μL was selected for subsequent experiments.

As a substrate, dNTPs play an important role in amplification. Excess dNTPs would bind to free magnesium ions in the reaction system, resulting in the inhibition of availability. When the concentration of dNTPs in the reaction system was excessive, it was more conducive for the amplification of PEDV, while the increment of PoRV and PBoV decreased significantly. When the concentration of dNTPs was 1.2 mM, the differences in the T/C values of the three targets were negligible (Figure 2C).

The appropriate temperature is conducive to stimulating the activity of *Bst* DNA polymerase. The amplification efficiency of the three target fragments was higher at 64 °C, and the expansion was similar. When the temperature exceeded 70 °C, it was not conducive to the reaction (Figure 2D). 

Another parameter, the reaction time, was optimized on the basis of previous experiments. With the extension of the amplification time, amplification products gradually accumulated. When the reaction time reached 30 min, all three T-lines were nearly saturated, and the T/C values were relatively similar (Figure 2E).

### 3.3. Testing of Specificity and Sensitivity for the Triplex LAMP–LFD Assay

When testing the primer specificity, the results showed that only the detection lines of the corresponding targets appeared as red bands (Figure 3A). Then, other porcine pathogens, including PCV1, TGEV, PRRSV, PRV, and PPV, were used to verify the specificity of the assay. No red bands were observed on the three T-lines. This indicated that no cross-reactions occurred among these viruses (Figure 3B).

The standard plasmids were used as templates to determine the sensitivity of the triplex LAMP–LFD assay, and the data were obtained for standard curves. It was found that the detection limits of PEDV, PoRV, and PBoV were 2.40 × 10^1^ copies/μL, 2.89 × 10^1^ copies/μL, and 2.52 × 10^1^ copies/μL, respectively (Figure 4). The sensitivity evaluation results of the triplex LAMP–LFD assay were consistent with the results of the single and duplex LAMP–LFD assay (Appendix A).

### 3.4. Field Sample Testing

In total, 125 animal feces samples were analyzed. The triplex LAMP–LFD assay was compared with the rt-qPCR assay to verify the reliability. The analyses showed that the proportion of mixed infection between viruses was higher (Figure 5). The detection results of the rt-qPCR assay were perfectly consistent with the results provided by the pig farm. There were eight (6.4%) and nine (7.2%) PEDV positive specimens detected by the triplex LAMP–LFD assay and the rt-qPCR assay, respectively, eight (6.4%) and eight (6.4%) for PoRV and seven (5.6%) and six (4.8%) for PBoV (Table 2). The coincidence rate between the triplex LAMP–LFD assay and the rt-qPCR assay was over 99% (Table 2), which demonstrated great consistency between the two methods. The prevalence (P), positive predictive value (PPV) and negative predictive value (NPV) for the two methods were 8.8%, 90.9% and 99.1%, respectively (Table 3). 

## 4. Discussion

Currently, a large variety of porcine pathogens have been found in diarrhea samples from pigs. Viral diarrheal diseases caused by porcine pathogens and the co-infection with multiple pathogens are very prevalent in piglets with diarrhea [1]. The harm caused by PEDV, PoRV, and PBoV mixed infection could not be ignored [4,30,31,32]. In order to prevent and control the spread of the diseases, some national standards for the testing of porcine diarrhea viruses were established in China, such as GB/T 36871-2018 (using a multiple RT-PCR assay to detect TGEV, PEDV, and PoRV) and GB/T 34757-2017 (using an RT-LAMP assay to detect PEDV) [51,52]. The simultaneous detection of different objects in a single reaction, represents a novel technical support approach for the rapid diagnosis of pathogens and the determination of epidemic etiology. The effective and rapid testing of multiple targets has become key to early detection for porcine viral diarrhea.

Contemporary, laboratory testing methods mainly include virus isolation, immunoassays and molecular biology assays [53]. The virus isolation method cannot meet the needs of rapid testing; the immunological methods require the preparation of the antigens and antibodies in advance, which would be greatly affected by the environment [38,39]. With the development of molecular biology technologies, diagnostic methods based on nucleic acid detection have gradually become an important technical means of animal epidemiological detection [53]. Derivative techniques based on PCR, such as RT-PCR, multiple PCR, and fluorescent quantitative PCR, have been used to detect these porcine pathogens (Appendix A). These methods are effective, highly sensitive and specific [53]. When conventional PCR was selected to identify the porcine pathogens, reactions could be completed within 80–90 min through the thermal cyclometer (Appendix A). However, the amplified products were not visible to the naked eye, and needed to be verified by means of agarose gel electrophoresis after the reaction. In order to satisfy the requirements of multi-target simultaneous detection and the visualization of detection results, qPCR was gradually applied. For instance, a real-time PCR assay based on multiple *Taqman* probes was established for simultaneously detecting PEDV, PDCoV, PToV, and SADs-CoV, which was performed using a Roche LightCycler^®^ 96 Instrument within 40 min, with a detection limit of 1 × 10^2^ copies/μL for each pathogen [54]. In addition, the emergence of the digital PCR provided a more accurate and sensitive means for the diagnosis of pathogens. The droplet digital PCR (ddPCR) method for detecting PEDV could be accomplished in 80 min through the thermal cycler C100 Touch and QX200 droplet generator, with a detection limit of 0.26 copies/μL, which was a 5.7-fold increase in sensitivity compared with that of real-time PCR [55]. However, the widespread use of the techniques is limited, due to their reliance on thermal, electrophoresis or advanced precision equipment [38]. Isothermal amplification methods alleviate these limitations. In recent years, the LAMP assay has become an effective tool for detecting a variety of pathogens. Some LAMP-based methods, such as RT-LAMP, fluorescent quantitative LAMP and LAMP–LFD, have been applied for the detection of swine diarrhea viruses (Appendix A). Compared with the PCR-based molecular biology methods, the LAMP assays can conduct reactions under a constant temperature for approximately 60 min, almost alleviating the restriction of experimental conditions. The combination of multiple LAMP assays and LFD not only meets the requirements of the simultaneous detection of multiple targets but also satisfies the demand of the visualization of experimental results. The multiple LAMP–LFD assay is expected to be an effective means for the clinical identification of pathogens in the future.

In this study, a triplex LAMP–LFD assay was established to simultaneously detect PEDV, PoRV, and PBoV. It showed high specificity and sensitivity in testing samples in the field. The rt-qPCR detection results were completely consistent with the results provided by the pig farm. Although there were slight differences between the testing results of the triplex LAMP–LFD assay and the rt-qPCR assay, the two methods also demonstrated great consistency. For sample No. 26, the detection result of PEDV was positive for the rt-qPCR method, and the judgment was obtained after 33 cycles, which was interpreted by its Ct value. Conversely, the result was negative for PEDV when it was tested using the triplex LAMP–LFD assay. This difference might be due to the relatively lower viral load. False-negative results can easily lead to missed detection. For sample No. 58, the triplex LAMP–LFD assay was positive for PBoV, whereas the rt-qPCR assay was negative. The thermal cycle exceeded 35 repeats before the result occurred. The false-positive result might have been caused by aerosol pollution, since the LAMP products were opened for testing to realize the visualization of the results. In terms of the positive predictive value (PPV) and negative predictive value (NPV), the two methods demonstrated good accuracy. In a further study, it would be necessary to achieve a lower detection limit of the triplex LAMP–LFD assay, to ensure the specificity and address the issue of false-positive results whenever possible.

The triplex LAMP–LFD assay developed in this study could simultaneously detect three target viruses within 30 min, with detection limits of 2.40 × 10^1^ copies/μL, 2.89 × 10^1^ copies/μL, and 2.52 × 10^1^ copies/μL, respectively. When PEDV and PCV2 were detected simultaneously by a two-phase LAMP–LFD method, amplification was completed in approximately 20 min, and the detection limits were 0.1 ng/µL and 0.246 ng/µL, respectively [56]. Through conversion, the detection limit of the triplex LAMP–LFD assay was lower. Although the detection objects of the triplex LAMP–LFD assay were inferior to those of the quadruple real-time quantitative PCR (qRT-PCR) assay, the qRT-PCR assay relied on precise temperature changes to carry out the reactions, while the triplex LAMP–LFD assay did not, making it more applicable for field detection or detection in areas with resource shortages [57]. In addition, the triplex LAMP–LFD assay was more cost-effective than the rt-qPCR assay. We estimated the cost of the rt-qPCR assay to be approximately USD 4.00 per virus, or USD 12.00 for three viruses. However, using the triplex LAMP–LFD assay, we estimated a cost of approximately USD 9.00 per reaction. In summary, due to its rapid, cost-effective and highly sensitive features, the triplex LAMP–LFD assay was better suited for rapidly diagnosing a variety of diseases in the field. The technique established in this study is expected to be widely available for the testing of porcine diarrhea viruses in clinical diagnoses.

## 5. Conclusions

In this study, a simple, rapid, sensitive, and specific triplex LAMP–LFD assay based on PEDV-*gp6*, PoRV-*vp6*, and PBoV-*vp1* was established for swine diarrhea virus. The portable and low-cost assay could be completed at 64 °C in 30 min. Using a miniature metal bath makes the assay easy to perform in the field. By combining this with lateral flow dipsticks, the results could be visualized directly. The triplex LAMP–LFD assay could meet the requirements for the on-site testing of field samples. In conclusion, the multiple diagnostic method is a great choice for assessing porcine viral diarrhea pathogens.

## Figures and Tables

**Figure 1 animals-13-01910-f001:**
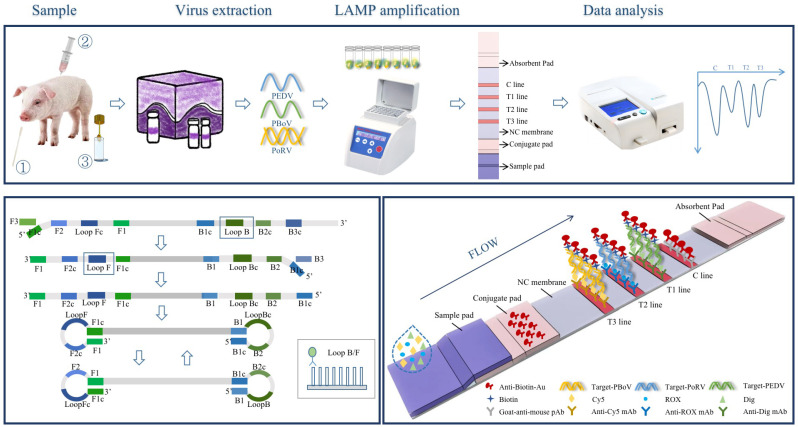
The schematic diagram of the triplex LAMP–LFD assay.

**Figure 2 animals-13-01910-f002:**
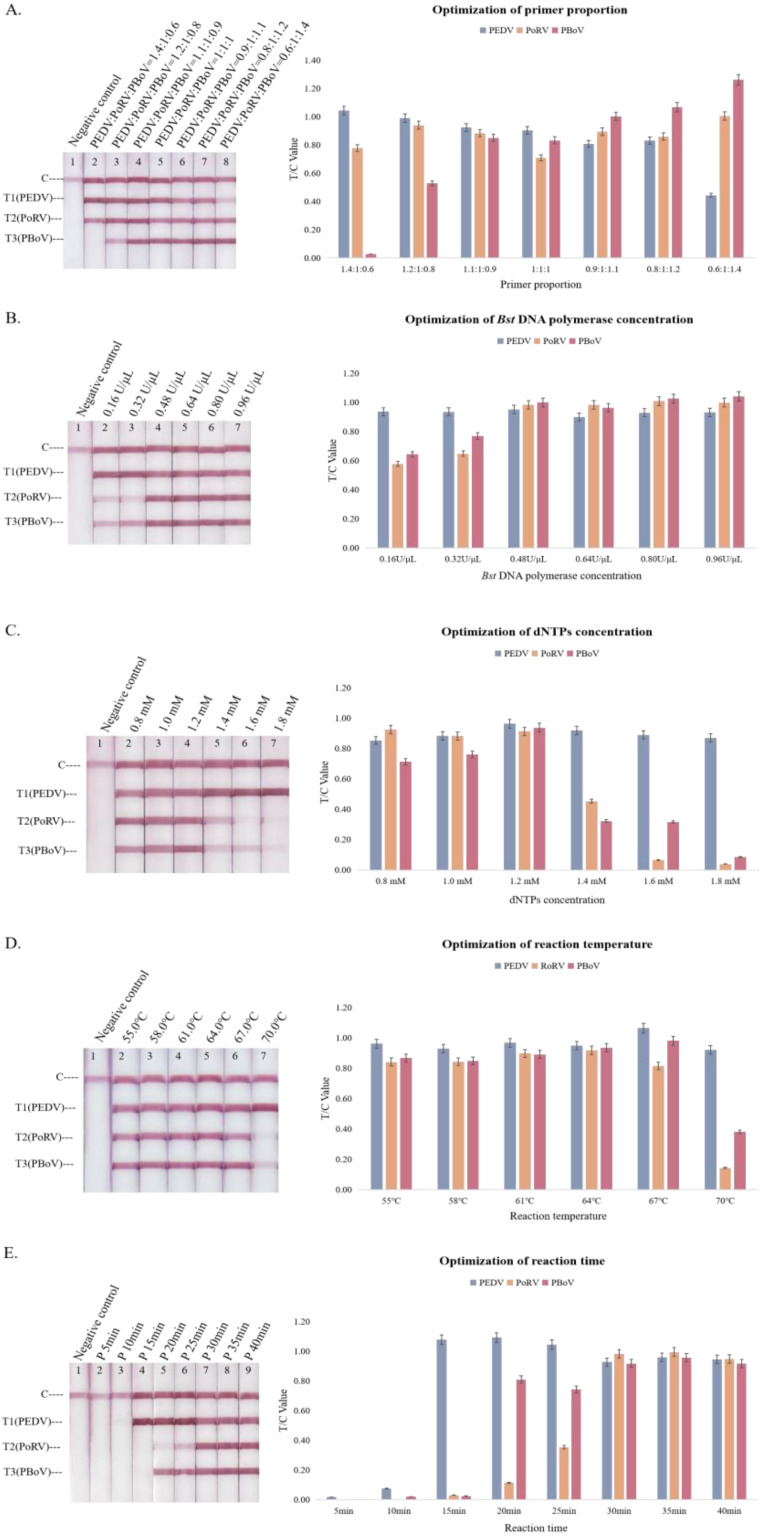
Optimization of the triplex LAMP–LFD reaction. (**A**) Primer mixtures ratios, (**B**) *Bst* DNA polymerase concentration, (**C**) dNTPs concentration, (**D**) reaction temperature, (**E**) reaction time. The data were collected using a GIC-S100-B14 test strip reader. The column represents T/C values.

**Figure 3 animals-13-01910-f003:**
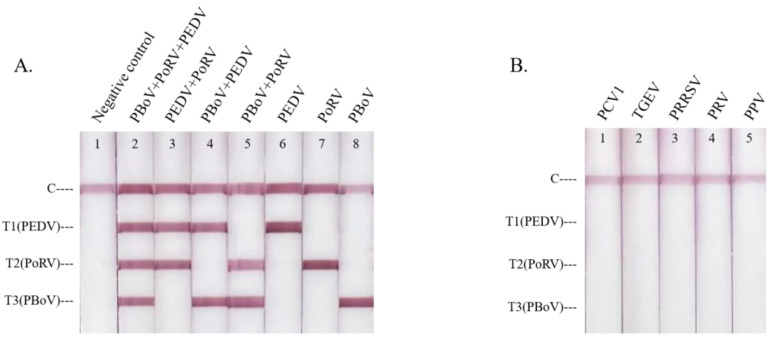
The specificity of the triplex LAMP–LFD assay. (**A**) Testing of three target pathogens, (**B**) testing of the other porcine pathogens.

**Figure 4 animals-13-01910-f004:**
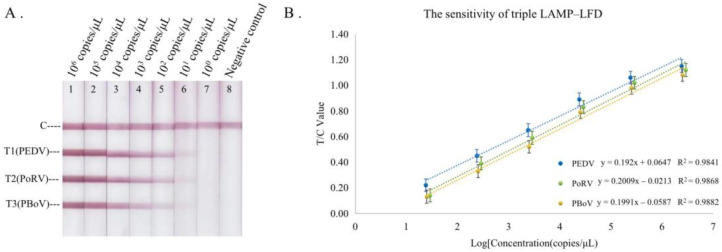
The sensitivity of the triplex LAMP–LFD assay. (**A**) Sensitivity testing, (**B**) standard curves. Using the logarithm of the plasmid copy number as the abscissa and the T/C value as the ordinate, the linear equation between the plasmid copy number and T/C value was obtained.

**Figure 5 animals-13-01910-f005:**
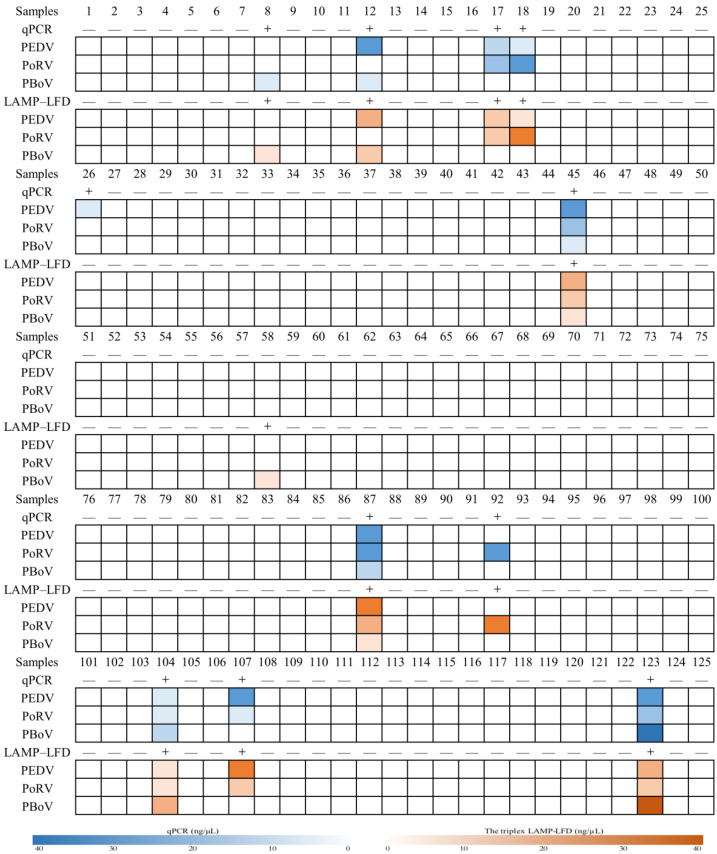
Field sample diagnostic results between the triplex LAMP–LFD and rt-qPCR assay. “+”, positive; “—”, negative.

**Table 1 animals-13-01910-t001:** Sequences of LAMP primers for detecting PEDV, PoRV and PBoV.

Target Gene	Primer	Sequence (5′–3′)	Modification	Length
PEDV-*gp6* NC_003436.1	F3	GGTACTTGCAAACAACGCTG	—	218 bp
	B3	TCTTTGCGCCTTCTTTAGCA	—	
	FIP	TCAATTCGCTCACCACGGCGTTTTCAAGGGGAATAAGGACCAGC	—	
	BIP	ACTACCTCGGAACAGGACCTCATTTTACCCAGAAAACACCCTCAGT	—	
	LF	AGCGAATTTGCTCATTCCAGTA	5′ Biotin	
	LB	GACCTCCGTTATAGGACTCGT	5′ Digoxin	
PBoV-*vp1* NC_023673.1	F3	CAACACCACAGTCGGGTAAC	—	228 bp
	B3	TTTCCCTCCCCCATCTGG	—	
	FIP	GCTCTGGACGCCAATTCTTGGTTTTTATTTACGCAACGGGACAAGT	—	
	BIP	GCAACAAGATGAGAGCCGACGTTTTTGGCATGGTTTCGTAGTAGCT	—	
	LF	TCCCATTCAATTTCGCAGGAG	5′ Biotin	
	LB	TACAAAATCAACGCCGATGGAGGAT	5′ Cy5	
PoRV-*vp6* KC113249.1	F3	CATGCTACTGTCGGACTT	—	198 bp
	B3	CAAGTTATCTTCTCTTGAAGGT	—	
	FIP	GCCGTTACATTTGCCAATAAAGTTTTTTTGAACTGAATCTGCAGTTTGT	—	
	BIP	TTCGTCAGGAATATGCTATACCAGTTTTTGAATAATTGGTAACCAGCTCTG	—	
	LF	CGTCCGCAAGCACAGATTC	5′ Biotin	
	LB	GACCAGTATTTCCACCAGGTATG	5′ ROX	

**Table 2 animals-13-01910-t002:** Consistency of the triplex LAMP–LFD detection and the rt-qPCR detection in specimens.

Test Result	Numbers of Samples (%) with rt-qPCR		Total Numbers of Samples (%)	Coincidence Rate (%)
	PEDV (+)	PEDV (−)		
LAMP–LFD ^a^				
PEDV (+)	8	0	8 (6.4)	99.2 (124/125)
PEDV (−)	1	116	117 (93.6)	
Total	9 (7.2)	116 (92.8)	125	
	Numbers of samples (%) with rt-qPCR			
	PoRV (+)	PoRV (–)		
LAMP–LFD				
PoRV (+)	8	0	8 (6.4)	100 (125/125)
PoRV (−)	0	117	117 (93.6)	
Total	8 (6.4)	117 (93.6)	125	
	Numbers of samples (%) with rt-qPCR			
	PBoV (+)	PBoV (−)		
LAMP–LFD				
PBoV (+)	6	1	7 (5.6)	99.2 (124/125)
PBoV (−)	0	118	118 (94.4)	
Total	6 (4.8)	119 (95.2)	125	

^a^ “+”, positive; “–”, negative.

**Table 3 animals-13-01910-t003:** The comparison of the accuracy for the triplex LAMP–LFD assay and the rt-qPCR assay.

Genotype	Numbers of Genotypes Found Positive by:			P ^a^ (%)	PPV (%)	NPV (%)
	LAMP–LFD	rt-qPCR	LAMP–LFD & rt-qPCR			
PEDV	8	9	8	8.8 (11/125)	90.9 (10/11)	99.1 (113/114)
PoRV	8	8	8			
PBoV	7	6	6			

^a^ “P”, “prevalence”; “PPV”, “positive predictive value”; “NPV”, “negative predictive value”.

## Data Availability

The data are contained within the article.

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
