# Peer review of "Triplex-Loop-Mediated Isothermal Amplification Combined with a Lateral Flow Immunoassay for the Simultaneous Detection of Three Pathogens of Porcine Viral Diarrhea Syndrome in Swine"

_animals, 2023, doi:10.3390/ani13121910_

Round 1
Reviewer 1 Report
I congratulate the authors for the work done. Overall the paper is well structured and the scientific data, information, and hypothesis are well addressed.
Estimating the financial advantage of using this POC method compared to the qPCR assay would be interesting. I suggest, if possible, adding some information to the discussion part.
A minor English review is necessary.
For example typo mistakes :
Line 313 : two paragraphs to end the main text should be deleted
Author Response
Dear reviewer #1:
Thank you very much for the time that you spent on reviewing our manuscript. Those comments are all valuable and helpful for revising and improving our paper. We have added some information to make the article more completed. The detail could be reviewed in the revised manuscript with correction red marked. We have improved the quality of our article language by professional English editing service of MDPI (english-edited-66488). All details can be seen in the revised manuscript with blue highlights.
Point 1: Estimating the financial advantage of using this POC method compared to the qPCR assay would be interesting. I suggest, if possible, adding some information to the discussion part.
Response 1: Thanks for the comment. According to the suggestion, we have calculated the cost of these two methods. It indicated that the triplex LAMP-LFD assay was more cost-effective than the qPCR assay. (Please see the revised manuscript, line 370-374)
Point 2: A minor English review is necessary. For example typo mistakes: Line 313: two paragraphs to end the main text should be deleted.
Response 2: Thanks for the comment. We have improved the quality of our article language by professional English editing service of MDPI (english-edited-66488). All details can be seen in the revised manuscript with blue highlights. In addition, we have deleted "two paragraphs to end the main text" from the manuscript and carefully checked to avoid these similar mistakes. (Please see the revised manuscript, line 386)
Reviewer 2 Report
This manuscript needs significant grammar/English edits and improvements.
Author Response
Dear reviewer #2:
Thank you very much for the time that you spent on reviewing our manuscript. Those comments are all valuable and helpful for revising and improving our paper. We have improved the quality of our article language by professional English editing service of MDPI (english-edited-66488). All details can be seen in the revised manuscript with blue highlights.
Point 1: This manuscript needs significant grammar/English edits and improvements. Response 1: Thanks for the comment. We have improved the quality of our article language by professional English editing service of MDPI (english-edited-66488). All details can be seen in the revised manuscript with blue highlights.
Reviewer 3 Report
Dear authors,
I am writing to you regarding the manuscript titled "Triplex loop-mediated isothermal amplification combined with lateral flow immunoassay for simultaneous detection of three pathogens of porcine viral diarrhoea syndrome in swine" (Animals-23100658).
The authors have successfully established a triplex loop-mediated isothermal amplification (LAMP) method combined with lateral flow dipstick (LFD) for simultaneous detection of Porcine epidemic diarrhoea virus (PEDV), porcine bocavirus (PBoV), and porcine rotavirus 21 (PoRV). They have designed LAMP primers targeting specific genes and demonstrated no cross-reactions with other porcine pathogens. Rapid, accurate, and cost-effective tests are urgently needed for early diagnosis, particularly for field screening. Therefore, the triplex LAMP-LFD assay described in this paper offers a promising solution for the detection of porcine viral diarrhoea pathogens. This assay has the potential to greatly facilitate on-site testing in field conditions, enabling prompt and targeted control measures to be implemented.
However, I believe that some revisions are required. Firstly, I suggest briefly discussing the economic impact of porcine viral diarrhoea in the introduction section. Secondly, you must include the STARD checklist and follow it to cover all elements of the list, including target and source population, sample size, blindness, etc. The data analysis section must also be improved. I think the manuscript would have greatly benefited from reporting the comparison of accuracy in terms of prevalence (P), positive predictive value (PPV), and negative predictive value (NPV) for the two methods. This would also improve Table 2.
Thirdly, the discussion section is too short. There are very few comparisons to previous molecular tests for the detection of the three target viruses. In my opinion, this section needs improvement before the publication of the manuscript to enhance the innovation of this detection method.
Additionally, extensive editing for typing mistakes and the English language is necessary throughout the manuscript. Typing mistakes indicate that little attention was paid before submission. Here are some examples: in line 110, "respectivly" should be corrected to "respectively"; in line 126, please add a space after 2.4 and remove the dot after "Preparation"; in line 160, please correct "st" to "at"; in lines 313-314, there is a sentence that has no meaning.
Best regards
Author Response
Dear reviewer #3:
Thank you very much for the time that you spent on reviewing our manuscript. Those comments are all valuable and helpful for revising and improving our paper. We have added all necessary information to clarify the findings and the details of tables. The details could be reviewed in the revised manuscript and supplementary materials with correction red marked. We have improved the quality of our article language by professional English editing service of MDPI (english-edited-66488). All details can be seen in the revised manuscript with blue highlights.
Point 1: However, I believe that some revisions are required. Firstly, I suggest briefly discussing the economic impact of porcine viral diarrhoea in the introduction section.
Response 1: Thanks for the comment. According to the suggestion, we have added two typical and large-scale outbreaks. On the one hand, the economic impact of porcine viral diarrhea came from the dead piglets, which were closely related to the number of piglet deaths and the unit price per piglet. On the other hand, the economic impact was reflected in the future prevention, control and disease treatment of pig farms, including vaccination and regular physical examinations. (Please see the revised manuscript, lines 43-50)
Point 2: Secondly, you must include the STARD checklist and follow it to cover all elements of the list, including target and source population, sample size, blindness, etc. The data analysis section must also be improved. I think the manuscript would have greatly benefited from reporting the comparison of accuracy in terms of prevalence (P), positive predictive value (PPV), and negative predictive value (NPV) for the two methods. This would also improve Table 2.
Response 2: Thanks for the comment. We have supplemented the STARD checklist according to your valuable suggestion. The checklist document was available online: https://www.equator-network.org.html (accessed on 17 May 2023), and it was been download to added all essential elements in supplementary materials. (Please see the revised Supplementary Materials #2)
In addition, we have supplemented table 2 and table 3 to improved the section of data analysis. The prevalence rate (P), positive predictive value (PPV), and negative predictive value (NPV), were added to demonstrate the accuracy for the triplex LAMP–LFD assay and the qPCR assay. (Please see the revised manuscript, line 277-285, Table 2 and Table 3)
Point 3: Thirdly, the discussion section is too short. There are very few comparisons to previous molecular tests for the detection of the three target viruses. In my opinion, this section needs improvement before the publication of the manuscript to enhance the innovation of this detection method.
Response 3: Thanks for the comment. We have supplemented and modified the discussion section according to your suggestions. We have compared the triplex LAMP-LFD assay with other molecular tests in terms of the cost, instrument dependence, reaction time, sensitivity, innovation and future prospects. (Please see the revised manuscript, line 314-331, 335-341, 366-378)
Point 4: Additionally, extensive editing for typing mistakes and the English language is necessary throughout the manuscript. Typing mistakes indicate that little attention was paid before submission. Here are some examples: in line 110, "respectivly" should be corrected to "respectively"; in line 126, please add a space after 2.4 and remove the dot after "Preparation"; in line 160, please correct "st" to "at"; in lines 313-314, there is a sentence that has no meaning.
Response 4: Thanks for the comment. We have improved the quality of our article language by professional English editing service of MDPI (english-edited-66488). All details can be seen in the revised manuscript with blue highlights. Specifically, we have corrected the typing mistakes, such as revised "respectivly" to "respectively", revised "st" to "at", added a space after "2.4" and remove the dot after "Preparation", deleted "two paragraphs to end the main text" in lines 313-314. In addition, we have checked the manuscript carefully to avoid these errors. (Please see the revised manuscript, line 126 , 143, 182, 386)
Round 2
Reviewer 3 Report
I am satisfied with the changes made by the authors. The paper looks much improved. I recommend the publication
Kind regards
Author Response
Thank you very much for the time that you spent on reviewing our manuscript. We do appreciate the comments from you that have helped us tremendously in revising the manuscript.